# Combined Role of Interleukin-15 Stimulated Natural Killer Cell-Derived Extracellular Vesicles and Carboplatin in Osimertinib-Resistant H1975 Lung Cancer Cells with EGFR Mutations

**DOI:** 10.3390/pharmaceutics16010083

**Published:** 2024-01-08

**Authors:** Aakash Nathani, Li Sun, Islauddin Khan, Mounika Aare, Arvind Bagde, Yan Li, Mandip Singh

**Affiliations:** 1College of Pharmacy and Pharmaceutical Sciences, Florida A&M University, Tallahassee, FL 32307, USA; aakash1.nathani@famu.edu (A.N.); islauddin.khan@famu.edu (I.K.); mounika1.aare@famu.edu (M.A.); arvind.bagde@famu.edu (A.B.); 2Department of Chemical and Biomedical Engineering, FAMU-FSU College of Engineering, Florida State University, Tallahassee, FL 32310, USA; li.sun@med.fsu.edu; 3Department of Biomedical Sciences, College of Medicine, Florida State University, Tallahassee, FL 32304, USA

**Keywords:** natural killer cells, extracellular vesicles, PD-L1, carboplatin, Osimertinib resistance, lung cancer

## Abstract

In this study, we evaluated IL-15 stimulated natural killer cell-derived EVs (NK-EVs) as therapeutic agents in vitro and in vivo in Osimertinib-resistant lung cancer (H1975R) with EGFR mutations (L858R) in combination with carboplatin (CBP). NK-EVs were isolated by ultracentrifugation and characterized by nanoparticle tracking analysis, and atomic force microscopy imaging revealed vesicles with a spherical form and sizes meeting the criteria of exosomal EVs. Further, Western blot studies demonstrated the presence of regular EV markers along with specific NK markers (perforin and granzyme). EVs were also characterized by proteomic analysis, which demonstrated that EVs had proteins for natural killer cell-mediated cytotoxicity (Granzyme B) and T cell activation (perforin and plastin-2). Gene oncology analysis showed that these differentially expressed proteins are involved in programmed cell death and positive regulation of cell death. Further, isolated NK-EVs were cytotoxic to H1975R cells in vitro in 2D and 3D cell cultures. CBP’s IC_50_ was reduced by approximately in 2D and 3D cell cultures when combined with NK-EVs. The EVs were then combined with CBP and administered by i.p. route to H1975R tumor xenografts, and a significant reduction in tumor volume in vivo was observed. Our findings show for the first time that NK-EVs target the PD-L1/PD-1 immunological checkpoint to induce apoptosis and anti-inflammatory response by downregulation of SOD2, PARP, BCL2, SET, NF-κB, and TGF-ß. The ability to isolate functional NK-EVs on a large scale and use them with platinum-based drugs may lead to new clinical applications. The results of the present study suggest the possibility of the combination of NK-cell-derived EVs and CBP as a viable immunochemotherapeutic strategy for resistant cancers.

## 1. Introduction

Cancer has continued to be one of the world’s top causes of death over the past half of the century, with a mortality rate second only to cardiovascular disorders [1]. The World Health Organization (WHO) estimates that cancer accounts for one in six global fatalities [2]. Every year, roughly 1.8 million new instances of lung cancer are identified, and the 5-year survival rate varies by region and stage, ranging from 4% to 17% [2,3]. Chemotherapy is still the most important treatment among all the strategies available, and the most effective chemotherapeutic drugs are synthetic drugs such as platinum-based regimens [3]. However, the major obstacle to the treatment is the significant side effects, such as cardiotoxicity, nausea, vomiting, hair loss, etc. [4]. Therefore, it is of utmost importance to understand and reduce the dose of these chemotherapeutic drugs while maintaining their efficacy.

The clinical viability of cancer immunotherapy, which stimulates the immune system to create anticancer effects, has been well studied [5]. Immunotherapy is now quickly developing and a clinically established treatment option for a variety of malignancies despite unsatisfactory early trial outcomes. Adoptive cell transfer, cancer vaccines, oncolytic viruses, and the delivery of antibodies or recombinant proteins are a few examples of immunotherapeutic techniques [6]. Adoptive transfer of immune cells, including natural killer (NK) cells, has become a focused approach to modulating the immune system against cancer [7]. NK cells are necessary for immunological monitoring against viral infections or tumors. It is well known that NK cells release a high quantity of interferon-γ (IFN-γ) to boost T helper type 1 (TH1) immune responses and also directly destroy tumor cells and cells infected with viruses [8,9]. On their cell surface, NK cells contain a variety of receptors that aid in differentiating between aberrant and healthy cells. Once they have identified their target cells, NK cells use diverse methods to kill them, including the release of tumor necrosis factor-α (TNF-α), perforin, granzymes, and Fas ligand (FasL) [10,11,12].

The majority of cell types release membrane vesicles of nanoscale called extracellular vesicles (EVs) that are known to contain a variety of cellular constituents such as proteins, DNA, mRNAs, and microRNAs (miRNAs) [13]. Receptor–ligand interactions and a variety of biochemicals, such as growth factors, hormones, cytokines, and chemokines, are involved in cell-to-cell communication. It is interesting to note that EVs have been shown to alter target cells’ phenotype by transferring their contents to distant target cells after the uptake [14,15,16]. EVs can, therefore, be used as innovative communication tools. EVs released by immune cells have the function of controlling both innate and acquired immune responses. Furthermore, current research indicates that EVs may affect immune responses that fight cancer [13]. Furthermore, it has been observed that natural killer cell-derived extracellular vesicles (NK-EVs), which include exosomes, exhibit cytotoxicity towards tumor cells and primary tumors that express a variety of NK-cell receptors [17,18,19,20]. Consequently, NK-EVs may prove helpful for immunotherapy; nevertheless, while proteins or miRNAs in the NK-EVs may be significant, the essential biological components in NK-EVs are still being investigated.

Carboplatin (CBP) (1,1-cyclobutyldicarboxylate) is a platinum-based anticancer agent that is frequently used and preferred over other platinum-based drugs because of its improved absorption and fewer adverse effects. It is specifically used to treat small cell lung cancer, head and neck cancers, and cancers of the testicles and ovaries [21]. Targeting the DNA directly, CBP conjugation is efficient in order to prevent transcription and replication, ultimately leading to cell death. The death or necrosis of tumor cells is caused by these DNA adducts, which have an effect on several transduction pathways [22]. CBP has been used in combination with antibodies, natural compounds, and other chemotherapeutic agents in therapies against various cancers [23,24,25,26]. Nano systems such as gelatin nanoparticles, chitosan nanoparticles, and targeted liposomes have been used to deliver CBP for better targeting and decreased side effects [23,27,28]. In our study, we used CBP in combination with NK-EVs to evaluate their potential in immunotherapy and to enhance CBP’s effect in resistant lung cancer.

Cytokine stimulation, which triggers the production of cytotoxic proteins, increases the cytotoxic activity of NK cells. However, it is still not known how cytokines affect NK-EVs or how target cells take them up [29,30]. One important cytokine that controls the growth and cytotoxic properties of NK cells is interleukin (IL)-15. It has been reported that IL-15 controls the JAK, PI3K, and MEK signaling pathways to support NK cell survival, proliferation, and cytotoxicity [31]. In this study, EVs were collected from NK cells that had previously been exposed to IL-15, and their biological properties, such as anticancer efficacy and capacity to target tumors using xenografts, were investigated in combination with CBP against Osimertinib (OSM)-resistant H1975 tumor xenografts. Further, the possible mechanism of the combination was also investigated.

## 2. Materials

NK92 cells obtained from American Type Culture Collection (Rockville, MD, USA) were cultured in MyeloCult H5100 media with horse serum, fetal bovine serum (FBS), and hydrocortisone from Stem Cell Technologies (Seattle, WA, USA). NK92 cells were transferred from a flask to a commercially available vertical wheel bioreactor (PBS biotech, Inc., Camarillo, CA, USA). CBP was acquired from Sigma-Aldrich (St. Louis, MO, USA) and was of good manufacturing practice (GMP) quality. Lung cancer cells (H1975) and RPMI-1640 medium were purchased from ATCC (Rockville, MD, USA). H1975 cells were cultured in RPMI-1640 medium complemented with 10% FBS and antibiotics (PSN; GIBCO, purcshased from Fisher scientific, Hanover Park, IL, USA). The cells were exposed to OSM to become resistant by using serial concentrations beginning from 0.5 nM to 10 nM over six months. FBS was purchased from Biotechne (Minneapolis, MN, USA). The cells were housed in a control humidified incubator at 37 °C with 5% CO_2_ and used within 15 passages. Sigma-Aldrich (St. Louis, MO, USA) provided all additional materials and reagents. Cell Signaling Technology provided all our research’s primary and secondary antibodies.

## 3. Methods

### 3.1. NK92 Cell Culture in 500 mL PBS-Vertical Wheel (VW) Bioreactor

NK92 cells were grown and extended up to passage 4 using culture media containing MyeloCult H5100 media with 12.5% horse serum, 12.5% FBS, and 5% fresh hydrocortisone. The cells were cultured and expanded in tissue culture flasks and placed in a typical incubator with 5% CO_2_. The media were changed twice a week. To obtain a single cell suspension for bioreactor culture, the cells were allowed to grow to 80–90% confluency. The cells were exposed to EV-depleted media the night before the inoculation into the bioreactor. The PBS-VW bioreactor was seeded at a density of 20,000 cells/mL. With the agitation speed set to 25 rpm, there were 15 min of static state and 5 min of agitation for 12 cycles, bringing the total time to 4 h. Cytokine IL-15 (25 ng/mL), which is well-known for enhancing the cytolytic activity of NK cells, was added to the media in the bioreactor [31]. Therefore, from this point, if otherwise mentioned, NK-EVs refer to EVs obtained from NK cells that are exposed to IL-15. The sampling of the bioreactor was performed every four days, and 0.5 mL of media was obtained from the PBS-VW bioreactor while it was agitated at 25 rpm. This was performed to gain an understanding of the metabolism of NK92 cells and identify the number of days after which the media must be changed. The levels of lactate and glucose concentration were evaluated for the spent media using a BioProfile Flex2 analyzer (Nova Biomedical, Waltham, MA, USA). Based on the metabolite results, forty percent of media were replaced every four days, and all the media were collected after two weeks [32,33].

### 3.2. NK-EV Isolation and Purification

The EVs were isolated from the spent media collected from the bioreactor using the differential ultracentrifugation method [34,35]. Subsequently, the EVs were characterized using nanoparticle tracking analysis (NTA). In brief, the media were filtered using a vacuum system with a pore size of 0.22 µm, followed by centrifugation at 500× *g* for 5 min at 4 °C. The supernatants were collected and subjected to a second centrifugation step at 2000× *g* for 10 min. Subsequently, the collected supernatants were subjected to another round of centrifugation at 10,000× *g* for 30 min and then underwent ultracentrifugation at 100,000× *g* for 2 h. Following the removal of the media, the pellets were subjected to a wash with phosphate-buffered saline (PBS) and subsequently subjected to an additional ultracentrifugation lasting 70 min. Following resuspension in 0.5 mL of PBS, the purified EV pellets were subjected to characterization using NTA and then finally stored at −80 °C for further experiments.

### 3.3. Characterization of NK-EVs

#### 3.3.1. Particle Size, Zeta Potential, and Concentration

The ZetaView^®^ BASIC NTA—Nanoparticle Tracking Video Microscope PMX-120, equipped with ZetaView software (version 8.05.11 SP4), was employed to determine the concentration of EVs in PBS suspension, particle size, and zeta potential. Prior to analysis, the isolated EVs were diluted with particle-free water at a dilution ratio of 1:1000. The assessment of the EVs was conducted in triplicate. The properties of NK-EVs were analyzed utilizing the dynamic light scattering (DLS) method at a temperature of 25 °C and a scattering angle of 90°. Following each sample reading, the laser chamber was thoroughly rinsed with particle-free water. The analysis of the data was performed using ZetaView Analysis software (version 8.05.12 SP1) [36,37].

#### 3.3.2. Atomic Force Microscopy (AFM)

AFM was used to investigate EVs that had been adsorbed by functionalization onto a mica sheet whose surface had been modified with 3-aminopropytriethoxysilane (APTES) and N,N-diisopropylethylamine (DIPEA) [38]. A Teflon sheet was affixed onto a fresh layer of muscovite mica sheet, which was subsequently positioned onto a metal specimen disc for AFM. Following the cleavage of the uppermost layer of mica, it was treated with a solution containing APTES and DIPEA for 2 h at 60 °C. In order to acquire a viable sample of EVs, the initial EV stock solution was diluted by a factor of 1000 using Milli Q water. A volume of 5 µL of the working solution was added to the mica surface. Following a 30 min duration, the mica sheet was subjected to a rinsing process using distilled water to eliminate any residual EVs. Subsequently, the provided specimen underwent peak-force quantitative nanomechanical mapping (PFQNM) analysis to investigate its response within a fluidic milieu, employing a Dimension Icon Scanasyst AFM. The attainment of highly accurate topographical details in the EVs was made possible with the utilization of a Scanasyst Fluid probe featuring a pyramidal tip. The probe’s pointed tip had a radius of 5 nm. The topography was evaluated by assessing the peak force of 300 pN and a scan rate of 0.1 Hz. The Nanoscope Analysis v1.9 application (Nanoscope Technologies, Bedford, TX, USA) was utilized to assess the morphological attributes, such as height and surface roughness, of the EVs. The determination of surface roughness was conducted by evaluating the root-mean-square deviation in feature height [37].

#### 3.3.3. Western Blotting of NK-EV Markers

NK-EVs and NK92 cell samples were lysed in radio-immunoprecipitation assay (RIPA) buffer (150 mM sodium chloride, 1.0% Triton X-100, 0.5% sodium deoxycholate, 0.1% SDS, 50 mM Tris, pH 8.0, and 1X Thermo Scientific Halt Protease Inhibitor Cocktail). The samples were then spun down at 14,000 rpm for 20 min after being lysed for 20 min on ice. Smith assay (Bicinchoninic acid assay) was used to measure the protein concentration after the supernatant was collected. The concentration of the protein lysate was standardized, and it was denatured for 5 min at 100 °C in 2X Laemmli Sample buffer. The Trans-Blot^®^ TurboTM Transfer System (Bio-Rad, Hercules, CA, USA) was used to electrophoretically transfer an equal concentration of protein from an SDS PAGE gel to a nitrocellulose membrane. The membrane was then blocked with PBS containing 0.1% Tween 20 and 3% *w*/*v* BSA for an additional hour at room temperature. Primary antibodies were added to the membrane at a dilution of 1:1000 in blocking buffer and incubated overnight at 4 °C. The blots were washed three times with PBS containing 0.1% Tween-20 (PBST) for 5 min each time, incubated for 1 h at room temperature with the appropriate horseradish peroxidase-conjugated secondary antibodies, and then washed three times with PBST. The Super Signal West Pico Chemiluminescent substrate was used to incubate the blots, and the Chemidoc Instrument (Bio-Rad) was used to take images. Using NIH ImageJ software (version 1.54), the immunoblots were measured using densitometry scanning [39,40].

#### 3.3.4. Proteomic Analysis of NK-EVs Followed by Gene Ontology Profiling

Protein extraction was carried out from both the NK-EVs and the parent cells (with three replicates for each group). According to the results of protein quantification, a maximum of 40 µg of proteins were extracted using the S-trap micro column (Protifi, K02-micro). The proteins were subjected to alkylation and digestion on a column according to the instructions provided by the manufacturer. The eluted peptides were fractionated using the Pierce high pH reverse phase peptide fractionation kit (Thermo Fisher Scientific, Grand Island, NY, USA, 84868), resulting in the division of each sample into five distinct fractions. Subsequently, the three replicates of the samples were subjected to vacuum drying and then sent to the Translational Science Laboratory at FSU for analysis using liquid chromatography–tandem mass spectrometry (LC-MS/MS). The samples were analyzed using the Thermo Q Exactive HF, following the previously stated methodology [41,42]. In summary, the raw files obtained were subjected to a search utilizing Proteome Discoverer 2.4, employing SequestHT, Mascot, and Amanda as the search engines. The Scaffold software (version 5.0) was used to verify the identification of the protein and peptide. The acceptance of peptide identification was based on the demonstration of a probability greater than 99.0% using the Scaffold Local false discovery rate (FDR) method. Similarly, the acceptance of protein identification was contingent upon the attainment of a probability threshold over 99.0% and the presence of at least two peptides that were recognized and acknowledged. The process of gene ontology (GO) annotation was performed using the g: Profiler tool.

### 3.4. Cell Viability Studies

#### 3.4.1. Two-Dimensional Cytotoxicity Assay

H1975-R cells resistant to OSM seeded in a 96-well plate (5 × 10^3^ cells/well) were treated with different concentrations of NK-EVs (0.25 × 10^11^–1 × 10^11^ particles/mL), CBP (10–500 μM), and combination for 48 h. After incubation, MTT assay was carried out as previously described [43].

#### 3.4.2. Magnetic Nanoshuttle 3D Cytotoxicity Assay

H1975-R cells were passaged and pelleted. Nanoshuttle solution (1 µL for 10,000 cells) was diluted in the media before being added to the pellet. The cell suspension was properly mixed and centrifuged three times at 800 rpm for 5–7 min. After centrifugation, the nanoshuttle solution-containing cell suspension was added to a 96-well plate with 12,000–15,000 cells per well. While seeding cells in the plate, a spheroid drive was used. The spheroid drive plate was incubated for three days at 37 °C, 5% CO_2_. On day 3, the media were changed. The plate was on the holding drive during the media change. After removing spent media, fresh media were added without mixing the cells. On day 5, the cells were treated for 48 h like 2D plates. After 48 h, the media in the wells were aspirated, and an MTT assay was performed [44,45].

#### 3.4.3. Dual Acridine Orange/Ethidium Bromide (AO/EB) Fluorescent Staining

The H1975-R cells were 3D plated using magnetic nanoshuttles at 12,000–15,000 cells per 96-well plate and maintained in a CO_2_ incubator at 37 °C for 24 h. NK-EVs, CBP, and combination were incubated for 48 h on 3D spheroid cells. Media were taken from wells after incubation, and cells were rinsed with PBS. Afterward, the cells were suspended in 200 μL of AO/EB staining solution, i.e., 10 μL acridine orange-AO at 100 µg/mL and 10 μL ethidium bromide-EB at 100 µg/mL diluted with PBS, and incubated for 10 min. After washing cells three times with PBS, a Nikon Eclipse Ti 100 inverted fluorescent microscope (Nikon Instruments, Inc., Melville, NY, USA) was used to take images immediately.

#### 3.4.4. Scratch Migration Assay

H1975 cells were dissociated using 0.25% Trypsin-EDTA (Sigma, Saint Louis, MO, USA) and seeded into a 6-well plate (5 × 10^5^ cells/well). The cells were grown to full confluence in RPMI 1640 medium for 24 h. Cell migration was analyzed using the Wound Healing Scratch assay. Using a sterile 200 μL tip, a fixed-width wound was created in the cell monolayer. Then, the cells were imaged at time 0. After imaging, the cells were washed with PBS three times to remove any residual cells. NK-EVs (1 × 10^11^ particles/mL), CBP (200 µM), and NK-EVs (1 × 10^11^ particles/mL) + CBP (200 µM) combination treatments were used. Images were captured after a 48 h treatment period using an Olympus microscope (CKX41) [32,39].

### 3.5. Cell Cycle Analysis by Flow Cytometry

H1975-R cells were cultured in 6-well plates at a density of 5 × 10^5^ cells per well and allowed to proliferate until reaching a confluency of 80–90%. The media were removed from the wells, and the cells were subjected to two washes with PBS. The cells were then treated with NK-EVs, CBP (100 µM), and a combination of both for a duration of 48 h. Following the completion of the treatment, the media were removed, and the cells were washed with PBS. After washing, trypsinization was performed using a 0.25% trypsin-EDTA solution. The resulting cell suspension was then subjected to centrifugation at 300× *g* for 5–7 min, after which the supernatant was carefully discarded. The cells were fixed in ice-cold 70% *v*/*v* ethanol for 30 min at 4 °C. The suspension was centrifuged at 300× *g* for 5–7 min. The cell pellet was reconstituted in PBS and subjected to centrifugation again. Then, the cells were treated with a solution of RNase A at a concentration of 100 µg/mL for 30 min. Then, the cells were resuspended in a staining buffer containing propidium iodide (PI) at a concentration of 50 µg/mL with a final volume of 1 mL. The suspension was incubated in the dark for 60 min at room temperature. Following this, the suspension was examined for the presence of DNA using the BD FACS Calibur flow cytometer (BD Bioscience Franklin Lakes, NJ, USA). The quantification of cell-cycle distribution in the G0/G1, S, and G2/M phases of the cell cycle was performed using Flow Jo software version 7.6.1. [32,46].

### 3.6. In Vivo Tumor Studies

BALB/c athymic nude mice (male, 6 weeks old, Foxn1^nu^) were purchased from Envigo. The American Association for Accreditation of Laboratory Animal Care approved institution provided the mice with housing and care under stringent pathogen-free guidelines. The animals were kept in typical housing cages with access to food and water. Animals were kept in normal enclosures at a temperature of 37 °C and a relative humidity of 60%. Florida A&M University’s Institutional Animal Care and Use Committee (IACUC) regulations were followed in all research (Protocol Number for animal studies: 023-02). Prior to the tumor studies, the animals had a week of acclimation.

In RPMI-1640 media supplemented with 10% FBS, H1975 cells (OSM-resistant) were grown under the recommended conditions of 5% CO_2_ and 37 °C in a controlled humidified incubator. H1975 cells (4 × 10^6^ cells) were mixed with Matrigel (1:1) and were then subcutaneously injected into the right flank of each mouse. The treatment was initiated ten days after the tumor cells were implanted. Five mice each were randomly assigned to one of four groups (untreated, NK-EVs alone, CBP, and NK-EVs + CBP combination). Every other day, the therapies (NK-EVs 200 µg, CBP 25 mg/kg, and combination) were given intraperitoneally. Using a digital Vernier caliper instrument, the tumor volume was measured throughout the course of 10 days of treatment. The tumor volumes were computed using the formula TV = ½ ab^2^, where “a” and “b” stand for the tumors’ length and width, and TV is the tumor volume [32,47].

### 3.7. Immunoblotting

Tumor tissue samples were homogenized by using T-PER^TM^ Tissue protein extraction reagent (cat no.78510) (Thermo Fisher scientific, Grand Island, NY, USA) containing protease inhibitor (cat no. P8340) and phosphatase inhibitor (cat no. P2850) (1:100) (Sigma Aldrich’s. Louis, MO, USA) by ice incubation for 40 min. The supernatant was collected after centrifugation at 14,000 rpm for 20 min. Protein concentration was determined by Bradford assay. Approximately 40–60 µg of protein was loaded on 4–12% SDS PAGE. The proteins were transferred to the nitrocellulose membrane, followed by blocking with 5% BSA in TBST for 2 h. Afterward, the membrane was kept on a shaker overnight with primary antibodies to PDL1, PD1, SOD2, PARP, SET, TGF- β, NF-kB, BCL2, and β-actin. The membranes were washed three times with TBST and then were incubated with horseradish peroxidase-conjugated secondary anti-rabbit and anti-mouse (1:3000) antibodies for 2 h at room temperature, and blots were developed by ECL (Bio-Rad, cat no-1705060) reagent on ChemiDoc^TM^ XRS+ Imaging system (Bio-Rad). Using densitometry and analysis software (Image J 1.36; Wayne Rasband, National Institutes of Health, MD, USA), the relative band densities were determined [48].

### 3.8. Statistical Analysis

All data were presented as the mean ± standard deviation (SD) of three replicates otherwise mentioned. By using either Student’s *t*-test for two-group comparisons or one-way ANOVA followed by “Bonferroni’s Multiple Comparison Test” for multiple variable comparisons in GraphPad Prism version 5.0 (La Jolla, CA, USA), the significance of the differences between the treatment groups was assessed. A *p*-value of less than 0.05 between the groups was regarded as statistically significant.

## 4. Results

### 4.1. NK Cell Growth and Expansion in PBS-Vertical Wheel Bioreactor

The EVs were isolated from NK cells grown in bioreactors and used for various in vitro and in vivo assays. NK cells were expanded in tissue culture-treated flasks and then seeded in a PBS-VW bioreactor. Media samples were measured for glucose and lactate concentrations. Active aerobic metabolism of cells was observed when grown in the PBS-VW bioreactors. Glucose concentration decreased from 5.10 to 1.21 and 0 mmol/L during days 0, 4, and 8, respectively, and lactate concentration increased from 2.34 to 6.66 and 8.66 mmol/L, respectively (Figure 1A). Interestingly, the molar lactate production/glucose consumption ratio was consistent at ~1.2 on day 4 and day 8, indicating that the energy metabolism of NK cells was aerobic in the PBS-VW bioreactor system. We also observed complete utilization of glucose in the medium by day 8. Other metabolites that supplement energy, i.e., glutamine, showed similar changes in concentrations in glutamine consumption and ammonium production, where more than half of the glutamine was utilized in 4 days (Figure 1B). Hence, it was decided to replace 40% media every 4 days in the bioreactor. Sodium, potassium, and calcium concentrations did not change significantly. Sodium and potassium concentrations increased on day 4 and then declined (Figure 1C,D), whereas calcium concentration decreased linearly (Figure 1E). Cell division and viability were higher in the bioreactor with IL-15 (Figure 1F,G).

### 4.2. Isolation and Characterization of NK-EVs

EVs were isolated from the media collected upon growing NK cells in the flasks and bioreactors through differential ultracentrifugation. The isolated EVs were characterized by NTA for average particle size concentration in terms of particle number per mL, as well as zeta potential (Figure 2A,B). The mean particle size of EVs collected from NK cells grown in 10 tissue culture flasks (182 cm^2^) to collect 500 mL media was found to be 101.2 ± 4.7 nm with an average particle number of 6.8 × 10^10^ particles/mL, and zeta potential of −18.98 ± 4.56 mV. The mean particle size of EVs isolated from medium collected from the bioreactor was found to be 89.5 ± 3.4 nm, with an average particle number of 2.2 × 10^11^ particles/mL with a 3.2-fold increase (Figure 1H) and zeta potential of −31.38 ± 0.25 mV.

AFM was employed to evaluate the topography of freshly cleaved mica surface as well as post-APTES: DIPEA treatment (Figure 2C,D). Height image of freshly cleaved mica surface and APTES: DIPEA treated surface showed no significant change in the topography. Representative images of NK-EVs shown at the nanostructure levels were used to evaluate the height profile and average surface roughness. NK-EVs displayed a height of 97.95  ±  11.82 nm, which was consistent with NTA data. Furthermore, the EVs obtained were smooth, with an average roughness of 2.84  ±  0.33 nm.

NK-EVs were isolated, and known exosomal markers were measured by Western blotting. The expression of TSG101, flotillin, alix, CD81, and CD63 was confirmed in each EV group, while the bioreactor group with IL-15 activated NK-EVs showed higher expression of perforin and granzyme B, which are responsible for NK cell-specific cytotoxicity. A negative marker calnexin was run in parallel to prove no cell debris contamination (Figure 2E).

### 4.3. Proteomics Analysis of NK-EV Protein Cargo

The protein cargo of the isolated NK-EVs was analyzed by liquid chromatography–tandem mass spectrometry (LC-MS/MS). Compared to control EVs that were isolated from NK cells without IL-15 exposure (control EV), NK-EVs that were exposed to IL-15 had 594 shared proteins and five distinct NK-EV-only proteins (Figure 3A). Compared to mesenchymal stem cell (MSC)-EVs, NK-EVs had 324 shared proteins (differentially expressed proteins-DEPs) and 178 distinct NK-EV-only proteins. For those distinct NK-EV-only proteins, a panel of miRNAs that are involved in regulating those proteins, such as miR-877, 615, 149, etc, was identified. (Figure 3B). Gene oncology (GO) analysis showed that these DEPs are involved in extracellular vesicles, programmed and positive regulation of cell death. The proteins that are involved in tumor cell cytotoxicity were identified, including proteins for natural killer cell-mediated cytotoxicity (Granzyme B) and T cell activation involved in immune response (plastin-2), T cell-mediated cytotoxicity (beta2 macroglobulin) and immune response to tumor cell (perforin-1), T cell-mediated immunity (myosin-Ig), T cell proliferation involved in immune response (40S ribosomal protein S6), and T-helper cell differentiation and inflammatory response and to antigenic stimulus, interferon-gamma-mediated signaling pathway, and type I interferon signaling pathway (high mobility group protein B1).

### 4.4. Effect of NK-EVs on Cell Viability and Migration

Cytotoxic effects of NK-EVs and CBP were assessed using viability assays wherein OSM-resistant H1975 R cells were exposed to different concentrations of NK-EVs and CBP for 48 h. NK-EVs showed cytotoxic effects in the cells with viability of 42.77% and 57.75% in 2D and 3D, respectively. However, IC50 values of CBP were found to be 195.2 ± 9.77 µM and 393.58 ± 11.26 µM in 2D and 3D cells, respectively (Table 1). In addition, we evaluated the efficacy of NK-EVs with CBP combination and observed that CBP showed an enhanced effect in 2D and 3D cells with IC50 values of CBP as 105.36 ± 5.45 µM and 212.80 ± 12.08 µM, respectively.

To visualize the cell viability by the distribution of live and dead cells in 3D after 48 h of incubation, the live/dead assay on H1975R cells was performed in 3D (Figure 4A,B). The cells remained alive in the control group for 24 h and developed a hypoxic core with dead cells after 48 h of incubation. Live and early apoptotic cells uptake the AO and emit green fluorescence, and late apoptotic and necrotic cells uptake EB and emit red fluorescence. The images show that with an increase in the concentration of CBP, there was a significant decrease in the number of live cells. Similar results were observed with the migration of cells where the combination treatment had the least number of migrated cells (Figure 4C). Overall, pictures depicted that NK-EVs and CBP had cell cytotoxicity, and the combination had the least relative mean cell viability (*** *p* < 0.001).

### 4.5. Cell Cycle Analysis

Flow cytometry of cell cycle analysis was performed after treating H1975-R cells with NK EVs, CBP, and NK EVs + CBP to study the distribution of cells in different phases (G1, S, and G2) and to detect the apoptotic cells with fractional DNA content (Figure 5). Both NK-EVs, CBP, and their combination significantly decreased the cell population in the G1 phase (from >83% to <60% in all treatments) and increased the cell population in the S phase (from <9% to >30% in all treatments)) when compared to control groups, indicating that the cells were undergoing cell cycle arrest in the S phase and the treatments were inducing apoptosis.

### 4.6. Effect of NK-EVs, CBP, and NK-EVs + CBP in OSM-Resistant H1975 Lung Cancer Tumor Model

BALB/c athymic nude mice (male, 6 weeks old, Foxn1nu) were used to study the effect of blank NK-EVs (≈2 × 10^11^ particles/mice or 200 µg; i.p), CBP (25 mg/kg; i.p), NK-EVs (2 × 10^11^ particles/mice or 200 µg; i.p) + CBP (25 mg/kg; i.p) on tumor volume at different days of treatment and were compared with control group (Figure 6). After the first and second days of treatment, there was no significant difference in tumor volumes of the different groups. However, on the fourth day of treatment, there was a slight difference in tumor volumes between the groups, and the difference was significant as compared to control for NK-EVs (* *p* < 0.05), CBP (** *p* < 0.01), and their combination (*** *p* < 0.001). Similarly, on the tenth day of treatment, all group treatments predominantly reduced the tumor burden in nude mice when compared to the control group. The combination worked the best among all the conditions in reducing the tumor burden. It was observed that NK-EV treatment alone had induced a significant reduction in tumor volume when compared to the control group. From the above findings, NK-EVs alone could reduce the tumor burden in the H1975 nude mice model, and the combination with CBP significantly promoted the antitumor effects.

### 4.7. Effect of NK-EVs, CBP, and NK-EVs + CBP on Apoptotic and Inflammation Markers in OSM-Resistant H1975 Lung Cancer Tumor Model

We evaluated the antiapoptotic effects of NK-EVs, CBP, and their combination in lung tumor tissues. In this study, we observed that PDL1, PD1, SOD2, PARP, SET, TGF-ß, NF-kB, and BCL2 were highly expressed in the control tumor tissue. Treatment with CBP reversed the expression of PDL1 (*p* < 0.001), PD1 (*p* < 0.001), SOD2 (*p* < 0.01), PARP (*p* < 0.01), SET (*p* < 0.01), TGF-ß (*p* < 0.01), NF-kB (*p* < 0.001), and BCL2 (*p* < 0.001), where NK-EVs showed a similar reduction in expression of PDL1 (*p* < 0.001), PD1 (*p* < 0.01), PARP (*p* < 0.001), SET (*p* < 0.01), TGF-ß (*p* < 0.01), NF-kB (*p* < 0.05) and BCL2 (*p* < 0.05), except for SOD2, where it showed no effect (Figure 7). The combination of NK-EVs + CBP further decreased the expression significantly (*p* < 0.001), except SOD2 (*p* < 0.01) (Figure 7). These findings imply that combining NK-EVs with CBP would be a superior therapeutic strategy in the treatment of OSM-resistant lung cancer in mice.

## 5. Discussion

Treating cancer using standard therapy methods such as chemotherapy, radiation therapy, and monoclonal antibodies is still very challenging. The World Health Organization estimated that malignant tumors caused 9.3 million deaths worldwide in 2021 alone [49]. As mentioned earlier, lung cancer remains one of the most lethal cancers in both men and women [50,51]. Despite years of research, the negative consequences of radiation and chemotherapy on patients’ immune systems and general health remain unresolved. Between 3 and 19% of patients refuse to receive chemotherapy or stop receiving it altogether [52]. These drawbacks highlight the need to design innovative cancer treatments that are highly targeted and efficient in killing cancer cells while sparing healthy cells. Hence, in this study, we combined a standard chemotherapeutic drug used in NSCLC, CBP with targeted immunotherapeutic particles (NK-EVs), to enhance their efficacy in lung cancer with minimal off-target side effects.

In the current study, differential ultracentrifugation was used to separate and purify NK-EVs from the NK-cell media. One of the primary obstacles to the therapeutic application of EVs is their low production yield from naturally secreted EVs. A PBS-VW bioreactor was used, which promoted NK-EV production with a cargo of increased clinical relevance while meeting the quantitative demand for EVs [53,54]. Richard et al. compared 2D culture of human umbilical cord mesenchymal stem cells (huMSCs) versus huMSCs on 3D microcarriers in a PBS-VW bioreactor and demonstrated that huMSCs altered the metabolic processes in the VW bioreactor microenvironment, increased EV yields, and promoted EV biogenesis markers. The agitation speed from 25 to 64 RPM did not influence the yield [33]. Hence, in this study, we maintained the agitation speed of 25 RPM throughout the cycle. Richard et al. also studied the metabolic activity of the cells at different agitation speeds by analyzing metabolite measurements in the media and found no substantial difference in glucose and glutamine uptake by the cells between 25 and 64 RPM. To understand the health of NK cells in the bioreactor environment, the spent media were tested for glucose, lactate, glutamine, and ammonium concentrations to evaluate cell carbohydrate and amino acid metabolism by measuring rates of glucose and glutamine consumption versus lactate and ammonium production, respectively. Glucose, a major source of energy, is converted to lactate via glycolysis by the cells grown in the culture [55,56]. At low glucose levels after day 4, net lactate generation from glucose decreased to zero by days 8 and 12. There was a steady flow of glucose into the cell mass, according to our complete examination of the lactate yield on glucose. It was observed that this rise was proportionate to glucose utilization but independent of lactate production. Similarly, glutamine is another metabolite that provides energy and is converted to ammonium. Glutamine to ammonium conversion decreased after day 4, suggesting that the cells used most of the media supplements in 4 days, and hence fresh media were added to the bioreactor every 4 days.

NK-EVs were found to have a size of less than 100 nm, which is the typical size of exosomes. Western blot analysis revealed that NK-EVs expressed alix, flotillin, TSG101, CD81, and CD63, all of which are exosome marker proteins [57,58]. Perforin and granzyme B, the typical proteins in NK-specific EVs, were also identified, which is consistent with previous studies using NK cell-derived EVs [19,59]. NK-EV morphology was examined by AFM, and NK-EV size, concentration, and zeta potential were measured by NTA. AFM has been previously employed to study the surface morphology of EVs such as salivary and MSC exosomes, etc. [60,61,62]. The size of NK-EVs measured by AFM was consistent with the value measured by NTA. Morphologically, AFM showed spherical vesicles with less surface roughness (less than 2 nm). AFM also allows the EVs to retain their shape during measurements, unlike other techniques like transmission electron microscopy (TEM), where dehydration of the EV membrane leads to a cup-shaped morphology and can mislead the data [63]. We demonstrate here that, while TEM is still the industry standard for analyzing EV morphology, combining NTA and AFM is a useful and better EV characterization strategy. The nanoscale NK-EVs can be used as passive therapeutic agents to treat a variety of solid malignancies. Cytokines have been used in cancer therapy because of their ability to induce anticancer immunity [64,65]. Young Kim et al. demonstrated the synergistic antitumor effects of NK-EVs stimulated with IL-15 and IL-21 (NK-exos^IL−15/21^) in Hep3B cells. Their findings revealed that NK-exos^IL−15/21^ expressed cytotoxic proteins (perforin and granzyme B) and also expressed exosome markers (CD9 and CD63). Further, NK-exos^IL−15/21^ induced the enhancement of cytotoxicity and apoptotic activity in Hep3B cells by activating the specific pro-apoptotic proteins (Bax, cleaved caspase 3, cleaved PARP, perforin, and granzyme B) and inhibiting the antiapoptotic protein (Bcl-2) [66]. Another study performed by Zhu et al. studied the NK-EVs, which were derived after exposure of IL-15 to NK cells. Their findings showed that NK-EVs_IL-15_ showed a significantly higher cytotoxic effect on human cancer cell lines (glioblastoma, breast cancer, and thyroid cancer) and increased the expression of molecules associated with NK-cell cytotoxicity. When compared with NK-EVs, NK-EVs_IL-15_ inhibited the growth of glioblastoma xenograft cells in mice significantly, which is the reason why we used IL-15 to stimulate NK cells [67].

Nanotechnology has been used successfully in delivering chemotherapeutic drugs in the last few decades [38,68,69,70,71,72,73]. However, EVs have been applied in diverse nano-biomedical fields, such as the application of unaltered EVs derived from immune cells for cancer treatment [19,74], as carriers for delivering chemotherapy drugs [75,76,77], and also using stem cell-derived EVs for treating ischemia and burn injuries [40,78]. Studies have shown that EVs produced from NK cells have the ability to eliminate tumors in cases of melanoma and glioblastoma [19,74]. NK cells release EVs that contain cytotoxic proteins and carry a variety of tumor-targeting receptors. This suggests that NK-EVs are capable of targeting tumor cells [79]. NK-EVs could potentially be included in the standard repertoire used by NK cells to eliminate cancer cells. These EVs possess a notable ability to extravasate and are likely to be less influenced by the hypoxic and acidic conditions within a tumor [80]. In our study, we aimed to examine the therapeutic efficacy of NK-EVs against Osimertinib-resistant H1975 cells by conducting in vitro assays and in vivo tumor xenografts. The proteins responsible for the anticancer effect of NK-EVs were identified using proteomics followed by gene ontology analysis and Western blotting. Previous studies have shown the presence of proteins involved in cell adhesion, cytotoxicity, and immune response [81,82]. Perforin and granzyme B are well-known effectors and were identified in the NK-EV proteome. They have been reported to have T cell and NK cell-mediated cytotoxicity [83,84,85]. Recent studies have provided evidence about the presence of perforin and granzyme B in NK-EVs [86]. The intrinsic pathway is activated by perforin, which also increases cytochrome-c release [87,88]. Cytochrome C is an electron-transport protein that is part of the respiratory chain and found in the mitochondrial intermembranous space [89]. Cytochrome C released from mitochondria associates with procaspase-9 in response to apoptotic stimulation [90]. Granzyme B has the highest apoptotic activity in the granzyme family due to its caspase-like ability to cleave substrates at aspartic acid residues, activating procaspases directly and cleaving downstream caspase substrates [91]. It prominently induces mitochondrial damage, which is a key feature of apoptosis [92].

One of the other proteins identified in our study was plastin-2, a member of the actin-binding protein family. Plastin-2 is involved in T cell activation in response to co-stimulation via TCR/CD3 and CD2 or CD28 [11]. It has been reported that plastin facilitates NKG2D-mediated NK cell migration and T cell activation [93]. Beta2-microglobulin and high mobility group protein B1 (HMGB1), which are directly or indirectly involved in T cell-mediated cytotoxicity or immunity, were also identified in our studies. Beta2-microglobulin can induce the expression of IL-6, 8, and 10 in a variety of cell types, regulate hormone/growth factor expression, and coordinate the interaction of cytokines and their receptors [94,95]. HMGB1, in eukaryotic cells, is a non-histone chromosome-binding protein that functions as an extracellular inflammatory cytokine. Extracellular HMGB1 acts as a damage-associated molecular pattern (DAMP) by binding to pattern recognition receptors (PRRs) as a chemokine or cytokine [96].

Myosin and 40s ribosomal protein S6 were identified in our study and are least likely to contribute to the observed anticancer effect of NK-EVs. Myosin from the actin family, for example, is a globular multifunctional protein family that is involved in many critical cellular processes such as muscle contraction, cell motility, cell division, vesicle formation, cell signaling, and the establishment and the preservation of cell junctions and cell shape [97]. In the case of cancer, abnormal expression and/or polymerization of the actin cytoskeleton have a significant impact on invasiveness and metastasis [98]. The findings from these experiments demonstrated that NK-EVs triggered apoptosis in tumor cells.

Because of CBP’s remarkable effectiveness against NSCLC tumors exhibiting EGFR mutations and its potential as a substitute treatment for malignancies resistant to OSM, we chose CBP as the therapeutic chemotherapy model for our study. CBP has demonstrated synergistic benefits in various tumors when examined in combination with other treatments in the past. In a phase III trial for recurrent ovarian cancer, a new therapy option of CBP-pegylated liposomal doxorubicin–bevacizumab was evaluated and standardized [99]. For older individuals with advanced non-small cell lung cancer, a phase II investigation validated the safety and effectiveness of combination chemotherapy with CBP and docetaxel [100]. Clinical data suggest that atezolizumab, when used with CBP chemotherapy, may be a viable treatment option with good outcomes for older lung cancer (small and non-small cell) patients who have not yet received treatment [101,102]. As such, CBP CBP-based combination with either chemotherapy or immunotherapy option may be considered for cancer patients. In our study, we chose to combine CBP with NK-EVs to have the advantages of both therapies.

NK-EVs’ anticancer efficacy was assessed using in vitro tests (i.e., live/dead assay in 3D, MTT assay in 2D and 3D cell cultures, scratch migration assay), which revealed that 48 h of NK-EV treatment reduced the number of tumor cells by 42% in 3D and 57% in 2D based on MTT assay. This outcome is in line with the live/dead cell viability test, which demonstrated that the treatment with NK-EVs reduced the viability of cancer cells. Furthermore, we observed that when the resistant H1975 cells were treated with NK-EVs, CBP, and their combination, there was a substantial difference in G1 phase arrest and an increase in the number of cells in the S phase as compared to control, indicating that the cells are going through apoptosis. The intrinsic apoptosis pathway, which is triggered by granzymes and perforin through enhanced apoptosome production, is activated by NK cells’ tumoricidal action [83,103]. In order to investigate the mechanism behind the cytotoxicity of NK-EVs against tumor cells in mice, Western blotting was used to determine the apoptosis-associated proteins. The results demonstrated that treatment with NK-EVs significantly downregulated the levels of PARP, BCL2, and SOD2 when administered alone and significantly decreased the levels when treated in combination with CBP. This suggested that NK-EVs triggered the pathways leading to apoptosis in tumor cells, thereby initiating cell death. According to the results of the current investigation, perforin and granzyme B were found in NK-EVs and present at significantly higher levels when primed with IL-15. This suggests that these proteins containing NK-EVs have the capacity to trigger the intrinsic apoptosis pathway [84,85].

Immunotherapy for cancer has entered a new phase since the discovery of therapeutics that can block certain immunological checkpoints, such as the cytotoxic T lymphocyte antigen-4 (CTLA-4) and, more recently, the programmed death-1 receptor (PD1) and its ligand (PDL1). Poor clinical outcomes have been connected to PDL1 expression in several human cancers, including hematological malignancies, lung cancers, and breast cancers. According to recent research, PDL1 expression doubles in tumor cells in NSCLC biopsies regardless of histology [104]. In addition, our investigations revealed that whilst both PD1 and PDL1 proteins were downregulated in the combo treatment, they were highly expressed in the control tumor samples. Targeting PD1 and PDL1, monoclonal antibodies have demonstrated promising results in NSCLC [105,106]. This implies that our combination’s unique method of targeting the PDL1/PD1 immune checkpoint axis also targets NSCLC through immunotherapy. These findings imply that NK-EVs caused cell death and that a combination of enhanced apoptosis and reduced tumor cell growth underlies NK-EVs’ anticancer effect. Tumor growth and progression are significantly influenced by inflammation [107]. Inhibiting inflammatory mediators, either specifically or non-specifically, improve survivorship by reducing the occurrence and progression of different tumors [108,109,110]. Many tumors are thought to respond well to treatment that targets inflammation [98,99]. We observed that the expression of NF-κB and TGF-ß proteins in tumor tissues was considerably reduced with NK-EV treatment or in combination with NK-EVs + CBP. Additionally, when granzyme A, a serine protease found in NK cells’ cytotoxic granules, is introduced into target cells, it causes cell death without the need for caspase [111]. Granzyme A has a specific target in the cell death pathway known as the SET complex, which is an endoplasmic reticulum-associated complex. Single-stranded DNA damage results from Granzyme A cleavage of SET, which relaxes the inhibition on NM23-H1 [112]. Granzyme A’s distinct death pathway offers a different approach for planned cell death, which is crucial for targets that are resistant to caspases. Our findings unequivocally show that NK-EVs can cause caspase-independent programmed cell death by breaking down SET protein. On the other hand, tumors can trigger endogenous immunological checkpoints and other immune evasion mechanisms. Although NK-EVs’ capacity to eliminate cancer cells and the associated processes have been investigated, more research is required to validate NK-EVs’ cytotoxic effects on primary tumors.

Two approaches have been proposed in the present clinical applications of EV research: (1) EVs as therapeutic agents in immunotherapy and (2) EV-mediated delivery. In this study, we focused on using EVs for their immunotherapy potential. However, EVs can also serve as drug delivery systems in cancer therapy [113]. The long-term potential of EVs in the marketplace is a big concern in academia and industry. Toward that goal, various investigators and pharmaceutical companies are scaling up exosome production using bioreactors. Various fiber-cell and wheel-based bioreactors are available, which can scale up to 5–10 L of the media and can be used to grow the cells [33,54]. Another aspect of exosome production is from milk. Milk exosomes can be easily scaled up since they are readily available from various animal sources like camel, cow, and goat [114,115]. Although combination therapy with CBP is known to work for treating cancers that develop resistance, there are drawbacks, such as adverse effects due to non-specificity and limited effectiveness, which prompted having a nanoparticulate delivery system for CBP [116,117]. As of now, CBP has been loaded into a number of nano drug delivery systems, including liposomes [118], nanoparticles [119,120], micelles [121], and carbon nanotubes [122], to show higher anticancer activity. The combination of CBP with Abraxane (Nanoparticle Albumin-Bound Paclitaxel) to treat small cell lung cancer significantly improved the efficacy, reduced the dosing frequency, and removed the requirement for weekly therapy as it showed anticancer activity in patients when it was given every 3 weeks [123]. The goal of having a combination therapy with a delivery system could be achieved by using NK-EVs. Therefore, our future work towards encapsulating CBP into NK-EVs will prove effective in not only enhancing the anticancer activity by their combination but also by channeling CBP to target the tumor and have a localized effect. Furthermore, EV-based clinical trials are expected to improve survival outcomes for oncological diseases in the near future, according to data from the National Institutes of Health’s clinical trials database [124,125,126,127]. During carcinogenesis, tumor cells damage their cell cycle checkpoints and are unable to stop the cell cycle for an extended length of time, which makes them resistant to chemotherapy. The best course of action to prevent this is to combine immunotherapy with chemotherapy [47]. Both in vitro and in vivo, NK-EVs demonstrated potent anticancer efficacy against resistant lung cancer cells. Additionally, these EVs can be loaded with additional anticancer drugs to help them pass through solid tumors or to increase their antitumor activity and tumor specificity. More research is needed to validate and improve the therapeutic efficacy of NK-EVs for treating incurable resistant lung cancer and to encourage the use of NK-EV therapy in clinical settings.

## 6. Conclusions

This is the first study to investigate the antitumor effect of NK-EVs, alone and in combination with CBP in drug-resistant lung cancer. NK-EVs significantly inhibited the growth of an OSM-resistant H1975 tumor xenograft when used in combination with CBP (1,1-cyclobutyldicarboxylate), a platinum-based anticancer drug. It was mainly due to the induction of apoptosis (SOD2 and PARP), anti-inflammatory response (NF-κB and TGF-ß), and its action on immune checkpoint pathways (PD1/PDL1). An exciting aspect of this study is that CBP and NK-EVs showed similar responses in tumor cell cytotoxicity in vitro and tumor regression in vivo, suggesting their role as an alternative to chemotherapy. Therefore, an immunochemotherapeutic treatment is the best way to deal with resistant cancers. Studies are ongoing with other tumor types, e.g., breast, colon, and gliomas, to investigate the role of NK-Evs further.

## Figures and Tables

**Figure 1 pharmaceutics-16-00083-f001:**
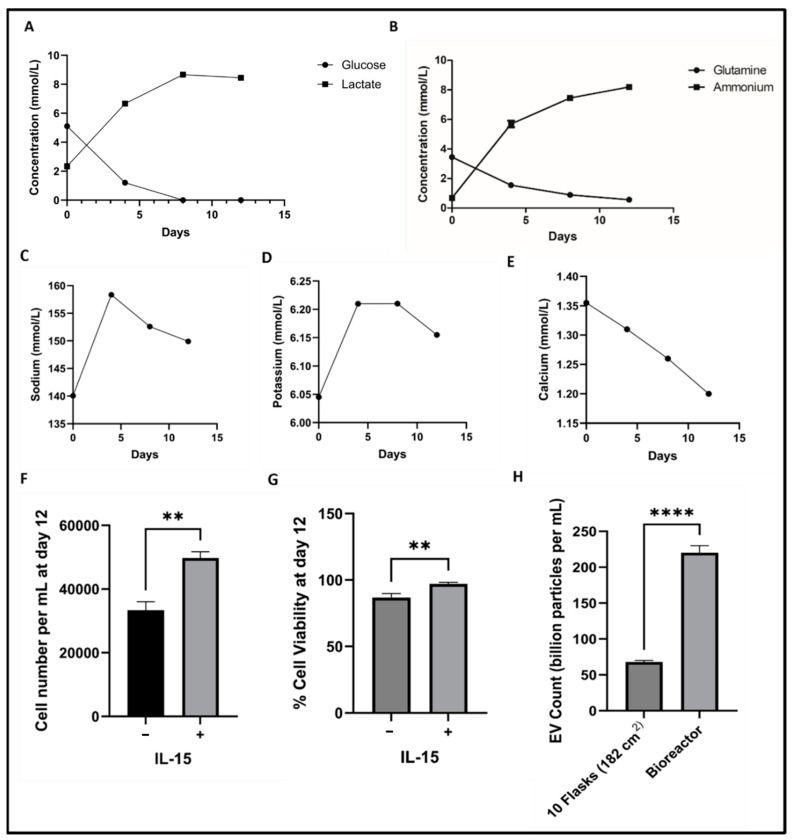
Metabolite measurements for NK cell metabolic activity in PBS vertical wheel bioreactor at 25 RPM from day 0 to 12: (**A**) Glucose and lactate concentration kinetics; (**B**) glutamine and ammonium concentration kinetics; (**C**) sodium kinetics; (**D**) potassium kinetics; (**E**) calcium kinetics; (**F**) cell number per mL on day 12; (**G**) cell viability on day 12 (**H**) NK-EV count from same volume of media collected from flasks and bioreactor. Data are shown as mean ± SD (*n* = 3). ** *p* < 0.01, **** *p* < 0.0001.

**Figure 2 pharmaceutics-16-00083-f002:**
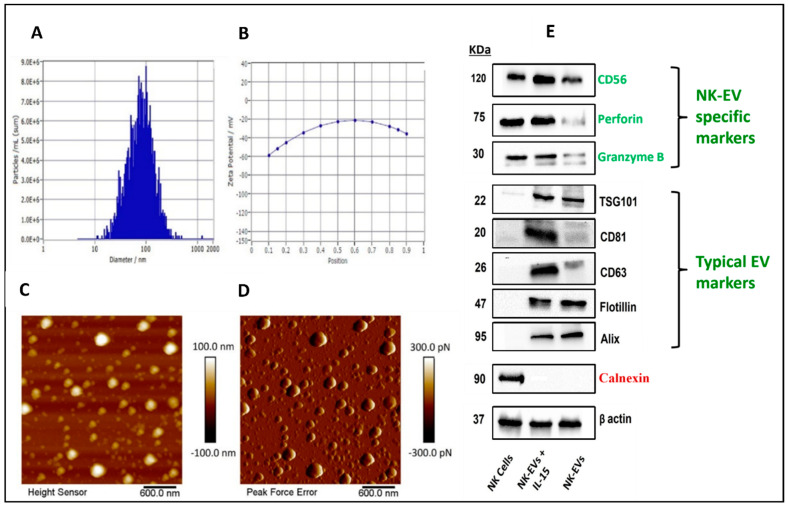
Nanoparticle tracking analysis of NK-EVs: (**A**) size and concentration; (**B**) zeta potential. Atomic force microscopy images of NK-Evs: (**C**) height image; (**D**) peak force error image; (**E**) Western blots of NK92 cells, NK-EVs + IL-15, and NK-EVs (without IL-15 exposure). Data are shown as mean ± SD (*n* = 3).

**Figure 3 pharmaceutics-16-00083-f003:**
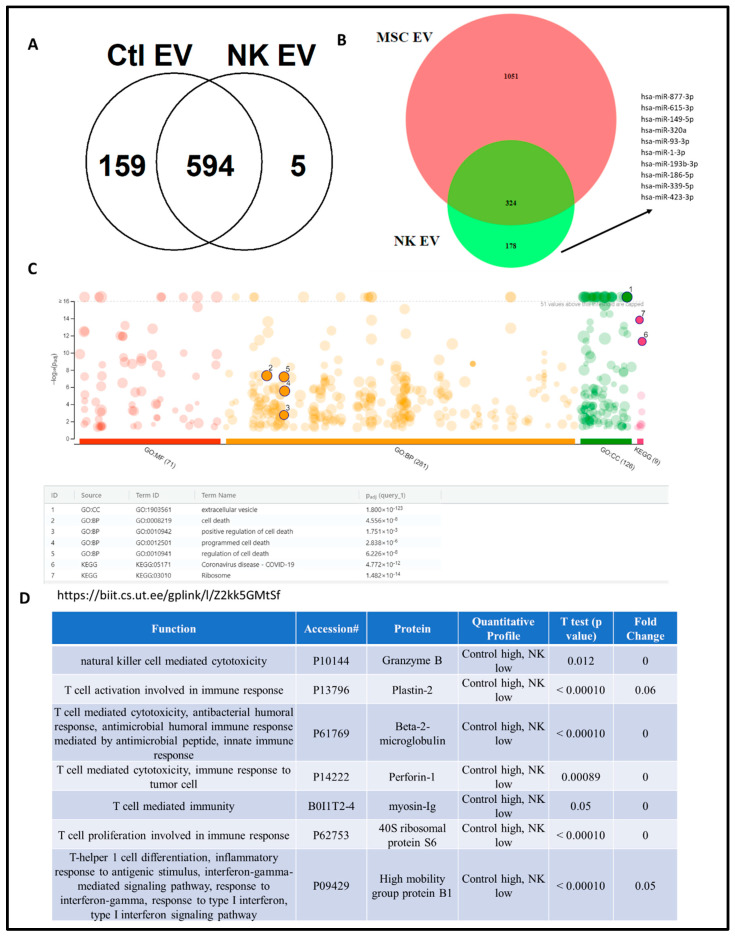
Proteomics of NK EV protein cargo analysis: (**A**) Venn diagram of NK-EVs compared to control EVs; (**B**) comparison of NK-EVs with MSC Evs; (**C**) gene oncology (GO) analysis of differentially expressed proteins (DEPs) for NK-EVs; (**D**) identified proteins in NK-EVs related to immune function.

**Figure 4 pharmaceutics-16-00083-f004:**
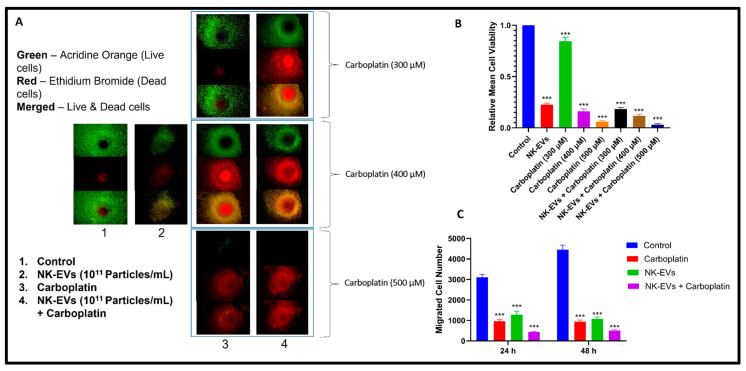
(**A**) Fluorescence microscopy images of live/dead assay: Green indicates live cells, and red indicates dead cells. The merged images indicate live and dead cells together. The images represent (1) control, (2) NK-EVs, (3) CBP (4) NK-EVs + CBP treatments against viability of OSM-resistant H1975 cell lines in 3D cultures. Scale bar = 200 µm. (**B**) The graph depicts relative mean cell viability obtained from live/dead assay. Migration assay of H1975-R cells in response to NK-EVs, CBP, and combination treatments: (**C**) Bar graphs represent NK-EVs, CBP, and NK-EVs + CBP treatments against number of migrated cells of H1975 cell line in 2D cultures. Data are shown as mean ± SD (*n* = 3). *** *p* < 0.001.

**Figure 5 pharmaceutics-16-00083-f005:**
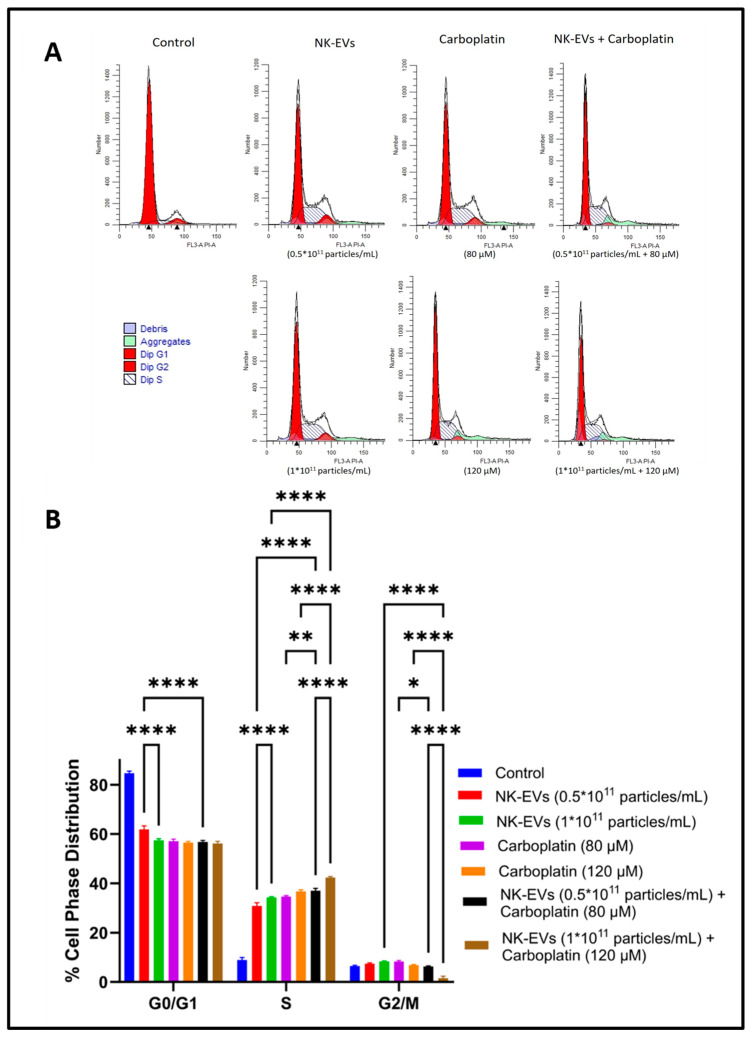
Cell cycle analysis by flow cytometry: (**A**) representation of the flow histograms control, NK-EVs, CBP, NK-EVs + CBP. (**B**) Bar graph showing cell cycle analysis after staining with propidium iodide (PI) in OSM-resistant H1975 cells treated with different treatments. All values are expressed as mean ± SD (*n* = 3). * *p* < 0.05, ** *p* < 0.01, **** *p* < 0.0001.

**Figure 6 pharmaceutics-16-00083-f006:**
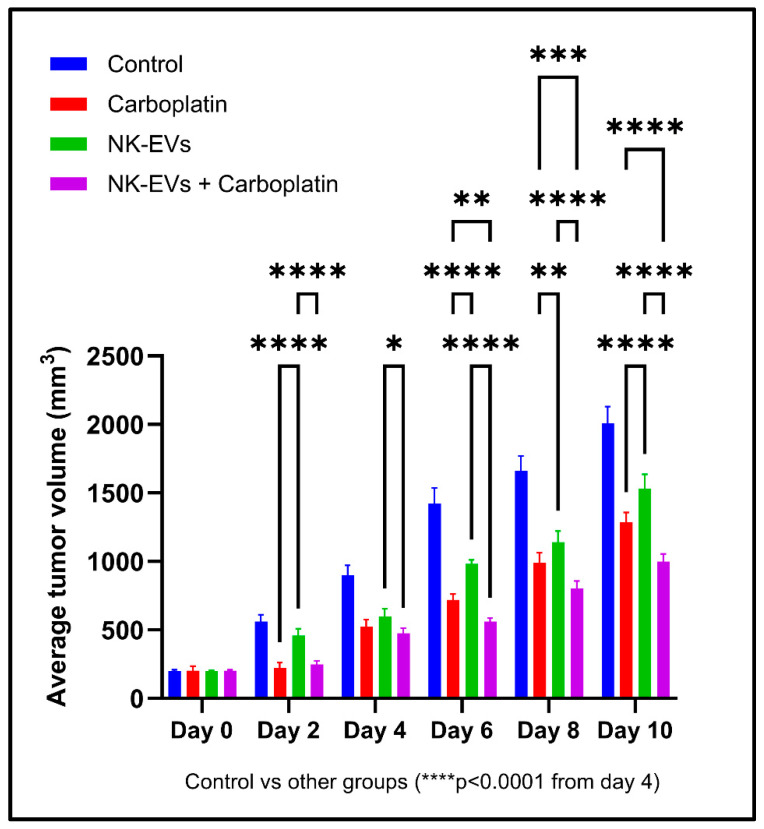
NK-EVs and CBP alone and in combination with each other reduced the tumor volumes in OSM-resistant H1975 lung tumor-bearing athymic nude mice. Bar graph represents the tumor volume data from day 0 to day 10 after starting the treatment with CBP, NK-EVs, and NK-EVs + CBP. Data are presented as mean ± SD (*n* = 5). * *p* < 0.05, ** *p* < 0.01, *** *p* < 0.001, **** *p* < 0.0001.

**Figure 7 pharmaceutics-16-00083-f007:**
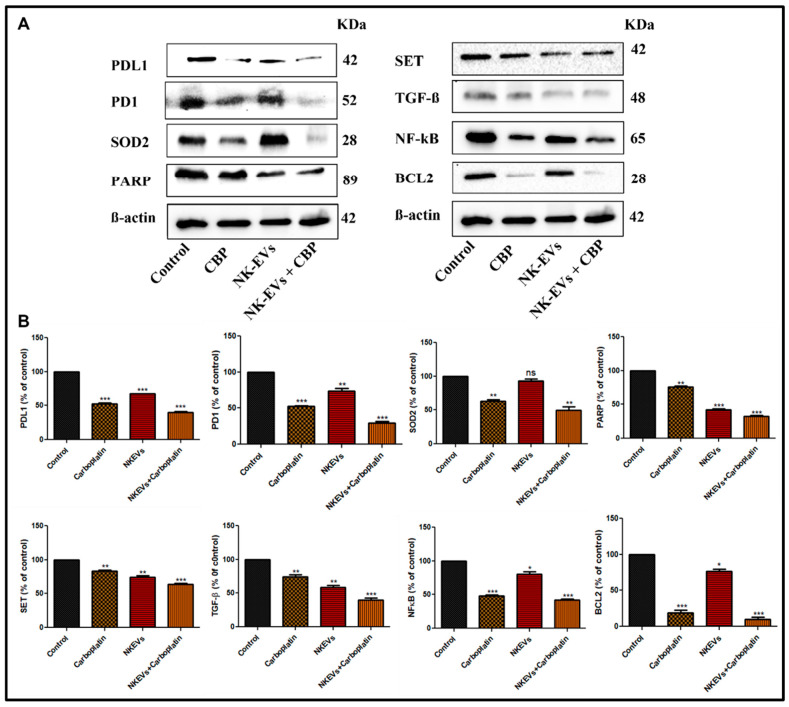
Effect of NK-EVs, CBP and their combination in tumors in BALB/c nude mice on apoptotic and inflammatory markers: (**A**) representative Western blots of CBP, NK-Evs, and NK-EVs + CBP; (**B**) blot quantification of respective PDL1, PD1, SOD2, PARP, SET, TGF-β, NF-kB, BCL2, and β-actin in the tumor tissue lysate represented by bar graphs. The values are expressed as mean ± SD (*n* = 3). ns is not significant. * *p* < 0.05, ** *p* < 0.01, and *** *p* < 0.001 vs. control.

**Table 1 pharmaceutics-16-00083-t001:** Half maximal inhibitory concentration (IC_50_) values CBP, alone and in combination with NK-EVs in OSM-resistant H1975 cell lines in 2D and 3D cultures.

Treatment	IC_50_ in 2D (µM)	IC_50_ in 3D (µM)
CBP	195.65 ± 8.12	383.64 ± 12.78
NK-EVs	55% Cell Viability	68% Cell Viability
CBP + NK-EVs	105.54 ± 4.98	226.12 ± 9.16

## Data Availability

The datasets generated during and/or analyzed during the current study are available from the corresponding authors upon reasonable request. The mass spectrometry proteomics data have been deposited to the ProteomeXchange Consortium via the PRIDE partner repository with the dataset identifier. Data are contained within the article.

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
