# Peer review of "Combined Role of Interleukin-15 Stimulated Natural Killer Cell-Derived Extracellular Vesicles and Carboplatin in Osimertinib-Resistant H1975 Lung Cancer Cells with EGFR Mutations"

_pharmaceutics, 2024, doi:10.3390/pharmaceutics16010083_

Round 1

Reviewer 1 Report

Comments and Suggestions for Authors

This manuscript presents a comprehensive study on the combined use of interleukin-15 stimulated natural killer cell-derived extracellular vesicles (NK-EVs) and carboplatin in targeting Osimertinib-resistant H1975 lung cancer cells with EGFR mutations. It explores the potential of NK-EVs, isolated through ultracentrifugation and characterized by nanoparticle tracking and atomic force microscopy, as therapeutic agents in both in vitro and in vivo settings. The study reveals that these NK-EVs, characterized by specific markers and proteins, exhibit significant cytotoxicity to resistant cancer cells and enhance the efficacy of carboplatin, a platinum-based chemotherapeutic drug. Additionally, the combination therapy shows promising results in reducing tumor volume in vivo and targets the PD-L1/PD-1 immunological checkpoint while inducing apoptosis and anti-inflammatory responses. This research suggests a new immunochemotherapeutic strategy for resistant cancers and paves the way for further studies in other tumor types.

To enhance the manuscript's clarity and impact, the following minor revisions are suggested:

Statistical Analysis (Section 2.8): Elaborate on the statistical methods used in the study. Clarify if the statistical data were presented as mean ± standard deviation (SD) of three independent experiments. Explicitly state the use of a Student’s t-test for two-group comparisons and one-way ANOVA with Bonferroni’s multiple comparisons test for analyses involving multiple variables.

Figure 1 - Kinetics of Glucose and Lactate: Address the inconsistency in the figure reference in the text (Figure 1A and 1B) and ensure it aligns with the actual figures. Also, rationalize the different units of measurement for glucose and lactate (g/L) versus other metabolites (mmol/L) in the illustration.

Figures 2 and 7 - Western Blot Images: Minimize the cropping of Western Blot images to provide a more comprehensive view of the experiments. Include and label molecular weight markers in all images. Clarify if internal controls or housekeeping proteins were processed on the same gel as the experimental samples.

Figures 5 and 6 - Consistency and Accuracy: Edit Figures 5 and 6 to have a consistent format. Address the discrepancy in Figure 6's description box, where it mentions data from day 1 to day 10, whereas the graph shows data from day 0 to day 10.

Method of Obtaining Vesicles (Line 144-145): Provide details about the pore size of the filter used for media filtration in the vesicle collection method.

IL-15 Content in Vesicles: Investigate whether the vesicles contain IL-15 post-isolation and its potential effects on tumor cells. Discuss the possibility that the observed effects of the vesicles could be partially attributed to the IL-15 content within them.

Discussion: Authors might consider referencing the findings from the study PMID: 33579033 to further enrich your discussion.

Reviewer 2 Report

Comments and Suggestions for Authors

Yan Li and Mandip Singh et al. reported an interesting work about lung cancer therapy. The topic was to some degree significance, and might arouse a certain impact in its field. The reviewer believed that a Major Revision is needed before a final acceptance. The detailed comments are as follows:

1.       The Abstract read a bit long. The reviewer supposed that some data less important could be removed in the Abstract.

2.       It was not necessary to showcase the details of centrifugation in the Graphical Abstract.

3.       The designed nanostructure of EV could be schematically illustrated at the end of the Introduction.

4.       Were there S.D. values in Figure 1A~E?

5.       Why not to show detailed IC50 values in Table 1 for NK-EVs?

6.       Please use unified fonts and typeface of words in all figures.

7.       The industrialization aspects of the EVs should be discussed in the Discussion.

8.       Please unify the format of Reference according to the guides of Pharmaceutics.

Round 2

Reviewer 2 Report

Comments and Suggestions for Authors

Thanks for your revision.